# Multiplex Analysis of Adipose-Derived Stem Cell (ASC) Immunophenotype Adaption to In Vitro Expansion

**DOI:** 10.3390/cells10020218

**Published:** 2021-01-22

**Authors:** Qiuyue Peng, Martyna Duda, Guoqiang Ren, Zongzhe Xuan, Cristian Pablo Pennisi, Simone Riis Porsborg, Trine Fink, Vladimir Zachar

**Affiliations:** Regenerative Medicine Group, Department of Health Science and Technology, Aalborg University, Fredrik Bajers Vej 3B, 9220 Aalborg, Denmark; qp@hst.aau.dk (Q.P.); mduda18@student.aau.dk (M.D.); gren@hst.aau.dk (G.R.); zxuan@hst.aau.dk (Z.X.); cpennisi@hst.aau.dk (C.P.P.); sriis@hst.aau.dk (S.R.P.); trinef@hst.aau.dk (T.F.)

**Keywords:** adipose-derived stem cells, heterogeneity, immunophenotype, cell subsets

## Abstract

In order to enhance the therapeutic potential, it is important that sufficient knowledge regarding the dynamic changes of adipose-derived stem cell (ASC) immunophenotypical and biological properties during in vitro growth is available. Consequently, we embarked on a study to follow the evolution of highly defined cell subsets from three unrelated donors in the course of eight passages on tissue culture polystyrene. The co-expression patterns were defined by panels encompassing seven and five cell surface markers, including CD34, CD146, CD166, CD200, CD248, CD271, and CD274 and CD29, CD31, CD36, CD201, and Stro-1, respectively. The analysis was performed using multichromatic flow cytometry. We observed a major paradigm shift, where the CD166-CD34^+^ combination which was found across all cell subsets early in the culture was replaced by the CD166^+^ phenotype as the population homogeneity increased with time. At all analysis points, the cultures were dominated by a few major clones that were highly prevalent in most of the donors. The selection process resulted in two predominant clones in the larger panel (CD166^+^CD34^−^CD146^−^CD271^−^ CD274^−^CD248^−^CD200^−^ and CD166^+^CD34^+^ CD146^−^CD271^−^CD274^−^CD248^−^CD200^−^) and one clone in the smaller panel (CD29^+^CD201^+^CD36^−^ Stro-1^−^ CD31^−^). The minor subsets, including CD166^+^CD34^−^CD146^−^CD271^+^CD274^−^CD248^−^CD200^−^ and CD166^+^CD34^+^CD146^+^CD271^−^CD274^−^CD248^−^CD200^−^, and CD29^+^CD201^−^CD36^−^Stro-1^−^CD31^−^, CD29^+^CD201^+^CD36^−^Stro-1^+^CD31^−^, and CD29^+^CD201^+^CD36^+^Stro-1^−^CD31^−^, in the seven and five marker panels, respectively, were, on the other, hand highly fluctuating and donor-dependent. The results demonstrate that only a limited number of phenotypical repertoires are possible in ASC cultures. Marked differences in their relative occurrence between distinct individuals underscore the need for potency standardization of different ASC preparation to improve the clinical outcome.

## 1. Introduction

Fat tissue is a rich source of cells, which, based on their multilineage differentiation potential, are referred to as adipose-derived stem cells (ASCs) [1,2]. During in vitro isolation from fat tissue by enzymatic dissociation and further processing, they are initially contained within a heterogeneous cell mixture termed stromal vascular fraction (SVF), together with other progenitor and more mature cell types. The potential for a positive impact on regenerative processes of crude SVF or populations more enriched for ASCs is great due to their significant pro-angiogenic, anti-apoptotic, pro-trophic, and immunomodulatory effects as well as their effect on extracellular matrix deposition and composition [3,4,5,6,7,8]. Currently, various ASC-based approaches are being assessed to treat some highly prevalent and recalcitrant conditions, such as osteoarthritis, cardiovascular disease, or chronic wounds [9,10,11,12,13]. However, it is important to note that these clinical trials employ either crude fat tissue, SVF, or primary cell cultures, which represent highly heterogeneous populations, also entailing cells that may not be active in supporting the required specific therapeutic outcome. It is undoubtedly of great interest to explore how the hallmark biological properties are distributed among the immunophenotypically discrete subpopulations, since such knowledge would be a major step towards cell-based therapeutics rationally designed for maximum efficacy.

Surface epitope expression patterns are commonly used to identify and select discrete subpopulations in order to determine their biological properties. Previous studies were able to determine the functional significance of some of the phenotypical variants; however, these investigations were based on the selection of either single markers or rather limited combinations thereof [14,15,16,17,18,19,20,21,22,23]. Based on the number and frequency of detectable surface epitopes, and their possible combinations, there appears to be a great number of distinct immunophenotypes within the ASC cultures [24,25]. Available evidence also indicates that the lineage heterogeneity is subject to substantial variation determined by the donor and culture-related factors [26,27]. To understand the impact of these factors on the potential clinical utility of different ASC preparations, the evolution of highly complex epitope patterns during the expansion in vitro should be elucidated.

We have previously examined the progression of more complex immunophenotypical changes during in vitro expansion on the tissue culture polystyrene surface in variants that were defined by a triple cluster of differentiation (CD) co-expression. Altogether, 15 surface epitopes were selected based on the potential significance with regards to mesenchymal stem cell (MSC)markers, wound healing, immune regulation, ASC markers, and differentiation capacity. For a more complete description of the rationale for the selection, please refer to the paper [28]. The epitopes were assorted into five particular combinations, CD90/CD105/CD73, CD166/CD271/CD248, CD29/CD200/CD274, CD146/CD34/CD31, and CD201/CD36/Stro-1. Among those, we found that the CD90/CD105/CD73 profile was invariantly present after expansion. Therefore, in the current study, we set out to identify stem cell subpopulations represented by their co-expression of the remaining 12 surface markers. To this end, two panels, enabling 7- or 5-fold multiplexing, were established and the cell subsets were identified using flow cytometry. This advanced approach allows for the detection of how highly defined ASC subpopulations appear and evolve during in vitro expansion.

## 2. Materials and Methods

### 2.1. Cell Isolation and Culture

Lipoaspirates were obtained after informed consent from three healthy donors undergoing elective liposuction surgery at the Aleris-Hamlet Private Hospital, Aalborg, Denmark. The protocols were approved by the regional committee on biomedical research ethics in Northern Jutland (N-20160025). Tissue collection complied with the principles defined by the Declaration of Helsinki and followed the rules defined by Danish legislation on anonymized tissue (Komitélov §14). The SVF was isolated as described previously [29]. Briefly, the aspirate fat, after being washed with sterile phosphate-buffered saline (PBS, Gibco, Taastrup, Denmark), was digested in Hanks’ Balanced Salt Solution (Gibco, Taastrup, Denmark) containing 0.6 U/mL collagenase NB 4 standard grade (Nordmark Biochemicals, Uetersen, Germany) on a shaker for 1 h at 37 °C. After digestion, the dissociated cells were referred to as a stromal vascular fraction (SVF). They were filtered through a 100 µm filter (Millipore, Omaha, NE, USA), and sedimented at 400× *g* for 10 min. SVF was then resuspended in prewarmed growth media (alpha-Minimum Essential Medium with GlutaMAX supplemented with 10% fetal calf serum and 1% antibiotics) (all from Gibco, Taastrup, Denmark), and clarified through a 60 µm filter (Millipore Omaha, NE, USA). The SVF was pelleted again by final centrifugation at 400× *g* for 10 min and resuspended in growth media. Nucleocounter NC-200 (Chemometec, Allerod, Denmark) was used to determine the cell yield.

The SVF suspension was plated into T175 culture flasks (Greiner Bio-one, Frickenhausen, Germany), and was referred to as passage 0 (P0). In total, three cultures were established from two donors and two cultures from one donor. The cells were released using TrypLE (Gibco, Taastrup, Denmark) when they reached 80–90% confluency (every 4–5 days), and the subsequent cultures were established at a density of 5000 cells/cm^2^. The population doublings (PD) were calculated using the formula PD = 3.32(log *Xe*−log *Xb*), where *Xe* is the cell number at the end of the passage and *Xb* is the cell number at the beginning of the passage [30]. On average, the population doubling during a passage was 1.65. The medium was changed twice a week. The cultures were propagated until P8.

### 2.2. Multichromatic Flow Cytometry (MFC)

To enable optimal simultaneous analysis, the markers were assorted into two panels, one entailing CD34, CD146, CD166, CD200, CD248, CD271, and CD274, and the other CD29, CD31, CD36, CD201, and Stro-1, as indicated in Table 1. The panels were designed to take into account the technical limitations of multichromatic flow cytometry. The combination of the specific antibodies and fluorophores, particularly for lowly expressed surface markers, were selected to minimize loss of resolution caused by spillover. In each run, only viable cells were considered with the aid of the Fixable Viability Stain 570 (FVS570) and all antibodies were directly conjugated (all from BD Biosciences, Lyngby, Denmark) as specified in Table A1. The particular staining and compensation steps followed essentially our previously published procedures [28]. Briefly, BD CompBeads Plus Set Anti-mouse Ig, ĸ and Anti-rat Ig, ĸ (BD Biosciences) were run first to determine the compensation values. The cell samples were incubated with FVS570 for 15 min at room temperature followed by the addition of antibody cocktails optimally diluted in BD Horizon Brilliant Stain Buffer (BD Biosciences) for 30 min at 4 °C. The working dilutions were determined by a series of titrations. In addition to the experiment samples, tubes omitting one of the antibodies were prepared for fluorescence minus one (FMO) controls.

The labeled cell samples were analyzed in a CytoFLEX (Beckman Coulter, Indianapolis, IN, USA), and the data analysis was done by the Kaluza 2.1 software package (Beckman coulter). The basic gate strategy and gate examples for the selected fractions can be found in Figure 1. Four basic gates were adopted to remove debris, ensure flow stability, discriminate doublets, and eliminate dead cells. The top 2.5 percentile of the FMO controls were utilized as the cutoff values to define the positive boundary, and the Boolean gates were employed to determine the potentially significant lineages whose proportions were over 5% at any passage.

### 2.3. Statistical Analysis

Statistical analysis was performed using IBM SPSS Statistics v.26 software package (IBM, Armonk, NY, USA). Data are shown as means of the means from two or three independent cultures from three unrelated donors + standard deviation (SD). Statistical comparisons were conducted using one-way repeated measures analysis of variance with Bonferroni post hoc tests. The significance value was set at 0.05.

## 3. Results

### 3.1. Phenotypical Variant Evolution

Panel A enabled analysis of the evolution of 128 possible cell subsets (Figure 2A). Upon invoking an inclusion criterion of average proportion from gated cells from three donors at any passage being higher than 5%, eight cell variants were identified. The CD combination profile of the selected subpopulations is listed in Table 2 and their frequencies of occurrence are specified in Table A2. Panel A demonstrates a prominent change taking place, whereby upon culturing, the four initially present cell subsets CSA1–4 disappear and two evolutionary rather distant and, at the same time dominant, variants CSA5 and CSA7 appear (Figure 2B,C). At the basis of this adaptation process is the loss of the CD34^+^CD166^−^ combination and acquisition of CD166 and its combinations, in particular with CD34, CD146, and CD271.

Panel B enabled analysis of the evolution of 32 possible cell subsets, and four significantly represented cell variants were identified (Figure 3A). The common denominator across all of these subsets is the expression of CD29. Contrary to panel A, a single subpopulation CSB2 (CD29^+^CD201^+^) dominates throughout the whole culture period, indicating that no major change is taking place (Figure 3B,C). However, less prominent, and opposing, trends can be discerned with the minor subsets. Here, the single CD29^+^ phenotype (CSB1) is detected only during the first four passages, whereas CD201 positivity along with CD36 (CSB4) or Stro-1 (CSB3) increases as a result of culturing.

### 3.2. Interpersonal Variability

As far as the subpopulations comprised in panel A are concerned, their occurrence differs markedly between different donors; nevertheless, the major variants, including CSA1–4, 5, and 7, appear to be expressed in a consistent fashion (Figure 4). In contrast, the minor subsets that appear during the course of expansion occur only sporadically, such as the CSA6 only at P4 in donor 3, and CSA8 only at P8 in donor 2. Similarly, in panel B, the dominant subset CSB2 is found in all donors, but what distinguishes it from panel A is the highly consistent prevalence rate at which it can be detected (63–87%). The minor subsets on the other hand appear highly irregularly, being apparently associated with a single donor, such as the CSB1 with donor 2, CSB3 with donor 3, and CSB4 with donor 1.

Additional insight into the clustering patterns was provided by multidimensional data rendering using radar spatial plots (Figure 5). In panel A, the radar plots indicate a shift from a broad P1 pattern involving four subsets, CSA1–4, towards two major subpopulations, CSA5 and 7, in P4 and P8 in all three donors. As explained above, this is taking place through the loss of the CD34^+^CD166^−^ phenotype and the acquisition of CD166 and combinations thereof. The radar plots, furthermore, reveal that the cell subsets cluster in donor-specific patterns. This stems from the fact that although the particular subsets from different individuals display an identical set of markers, they are expressed at relatively different levels. In a similar way, the panel B radar plots confirm a steady temporal pattern dominated by the CSB2, and further highlights the distinctiveness of each donor’s subset distributions.

## 4. Discussion

The single cell preparations resulting from the fat tissue enzymatic processing are inherently highly heterogeneous, harboring many different cell types, which upon in vitro culturing undergo adaptation and natural selection. These processes are reflected by an evolution of the immunophenotypical profiles of the expanded cells. To be able to fully harness the biological potential of ASCs, it is important to understand whether such changes bear any implications for the functional properties. Previously, it has been demonstrated that single surface markers are associated with specific functionalities, such as differentiation, angiogenesis, or bioactive secretion [14,15,31,32]. The single markers, however, provide only limited information about the clonal complexity. There is initial support for the notion that highly defined cell subsets are involved in enhanced specific functionality [24]; thus, a better understanding of the behavior of complex co-expression patterns would go a long way towards a better definition of the potency of different ASC therapeutic modalities [16,19,33]. A rational and effective application of ASC-based therapies could, thus, become a reality in spite of hindrances caused by a significant donor variability [34], differences in the site of the tissue of origin [35], or manufacturing processes [34].

One of the major observations of the currents study is the finding that although all 12 of the selected CD markers have been identified in the cultured ASCs [25,28], only some will participate in meaningfully represented variants. These comprise eight epitopes (CD166, CD34, CD146, CD271, CD29, CD201, CD36, and Stro-1), which altogether define 12 dominant variants in two different CD profile panels. On the other hand, CD274, CD248, CD200, and CD31 were found not to participate in any significant subpopulation, as they were distributed broadly among the great number of different marker combinations. In general, the cultures are dominated by few subsets, which are shared in all donors, but their relative proportions are highly individual, and this may carry non-negligible consequences for the clinical outcome of any given preparation. In contrast, the minor subsets are highly donor-dependent, but if they turn out to be associated with a specific functionality, they may also contribute to inter-individual differences regarding the therapeutic efficacy.

When looking for explanations for the temporal changes, two major mechanisms might explain why the proportion of distinct subpopulations changes with passaging; either differences in proliferative capacity between subsets or appearance of new phenotypes due to changes in expression of epitopes. In light of the de novo appearance of subsets, it can be concluded that while differences in proliferation rate may play a role, the major driving force of the phenotypic changes appears to be differential expression of surface markers.

From a temporal perspective, a remarkable switch is taking place, where the universal CD34 expression at the initial period is complemented with CD166 in later passages in accordance with our former observations [36]. CD166 has previously been reported, along with the other canonical markers CD73, CD90, and CD105, to indicate stem/progenitor cells [23,36]; thus, its appearance may have important connotations regarding the stemness potential of the propagated cell population. On the other hand, the rearrangement of CD34 from being a pan-clone hallmark into a single and novel CD166^+^CD34^+^ subset looks to suggest adaptation responses that underlie processes involved in the commitment [37]. It appears worthwhile that this variant be explored in more detail, while it seems to impart functionality in at least some of regenerative applications [38,39]. Another marker of interest is CD201, since together with CD29, it supports a highly prevalent and stable subset. To date, there is an opacity of evidence as to the role of this epitope within the context of human ASCs, but the notion that this particular clone supports hitherto unknown specific functionality warrants further investigation.

The current study provides an insight into the complex patterns of ASC subsets during adaptation to in vitro conditions. While identification of the major subsets of cell surface markers represents an important tool for the characterization of the cell product in a clinical setting, immunophenotyping alone cannot fully reveal the potency of an ASC population. The ensuing effort should aim to further associate the comprehensive surface epitope profile with biological potency. To this end, the major as well as minor lineages need to be separated and comprehensively evaluated in functional assays. These investigations will undoubtedly contribute to improving the ASC therapeutic value in a wide range of regenerative applications.

## Figures and Tables

**Figure 1 cells-10-00218-f001:**
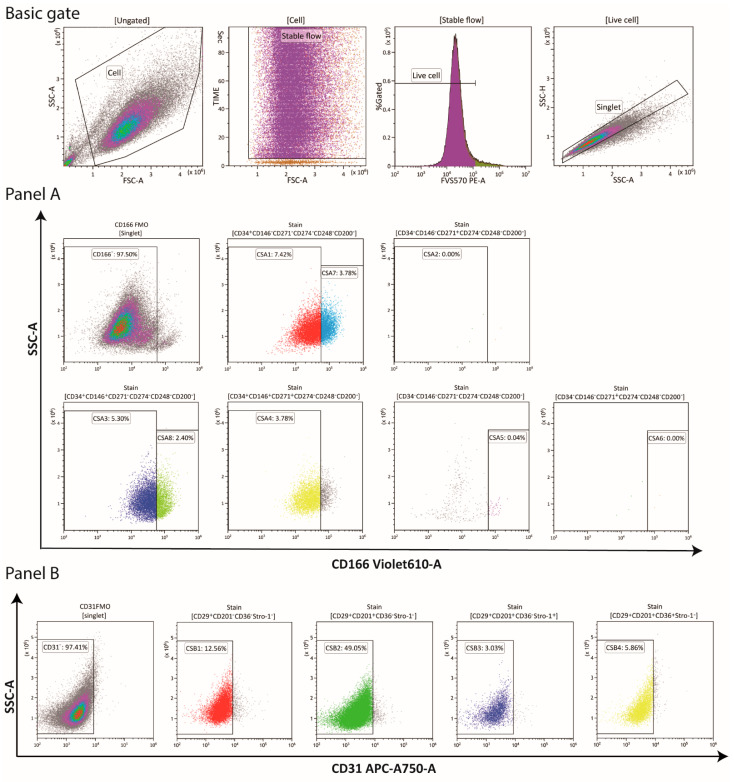
Flow cytometric and gating strategy. Four basic gates, one (FSC-A vs. SSC-A) to discriminate cells from debris, one (FSC-A vs. time) to ensure flow stability, one (viability intensity histogram) to exclude dead cells, one (FSC-A vs. FSC-H) to collect singlet cells, were used to target the analyzed cells. Representative gating strategies are demonstrated for the seven and five markers co-expressing variants in the panel A and B, respectively. These subpopulations were gated in colors using FMO controls and Boolean gates. The top 2.5 percentile of FMO control was set to define the positive boundary of the antigen intensity of the *x*-axis. Abbreviations: FSC-A, forward scatter-area; FSC-H, forward scatter-height; SSC-A, side scatter area; FMO, Fluorescence minus one; CSA, cell subset panel A; CSB, cell subset panel B. **^−^**, absence of the marker; **^+^**, presence of the marker.

**Figure 2 cells-10-00218-f002:**
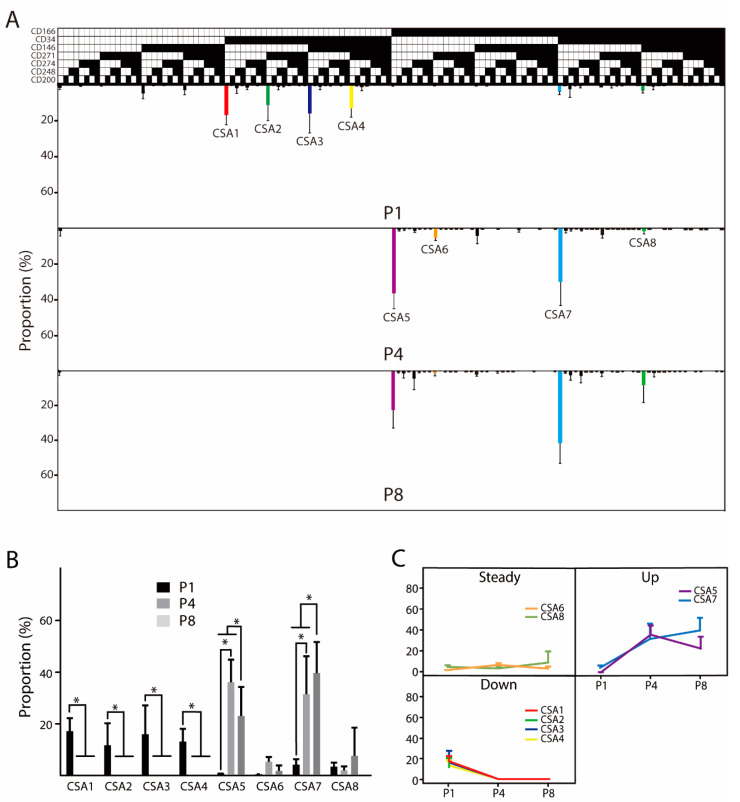
Evolution in the culture of ASC immunophenotype variants in panel A providing for combinations of seven cell surface markers. (**A**) The temporal changes in the distribution of distinct cell subsets are visualized using a tree plot. (**B**) Quantitative analysis of evolution trends. Only cell subsets with the relative prevalence of more than 5% at any passage are accounted for. (**C**) Subpopulation adaptation trends. The data represent the means of the means from two or three independent cultures from three unrelated donors + standard deviation (SD). The presence of a specific epitope is indicated by ∎, the absence by ⎕. Abbreviations: P, passage; CSA, cell subset panel A. *****, indicates a statistically significant change *p* < 0.05.

**Figure 3 cells-10-00218-f003:**
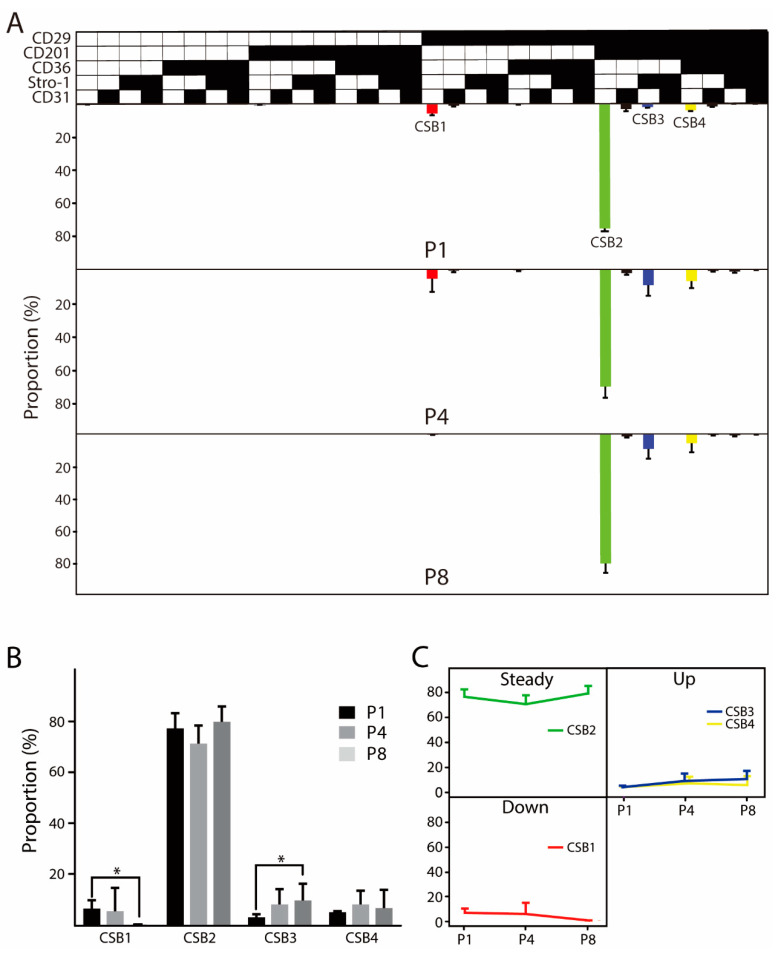
Evolution in the culture of ASC immunophenotype variants in panel B providing for combinations of five cell surface markers. (**A**) The temporal changes in the distribution of distinct cell subsets are visualized using a tree plot. (**B**) Quantitative analysis of evolution trends. Only cell subsets with the relative prevalence of more than 5% at any passage are accounted for. (**C**) Subpopulation adaptation trends. The data represent the means of the means from two or three independent cultures from three unrelated donors + standard deviation (SD). The presence of a specific epitope is indicated by ∎, and the absence by ⎕. Abbreviations: P, passage; CSB, cell subset panel B. *****, indicates a statistically significant change *p* < 0.05.

**Figure 4 cells-10-00218-f004:**
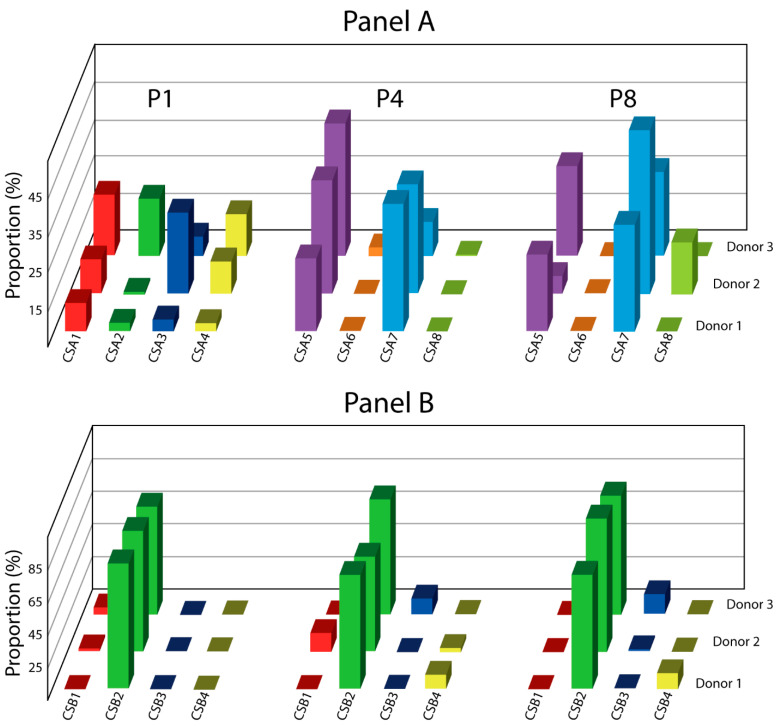
Evolution in culture of ASC immunophenotype variants from three unrelated donors. Only cell subsets with the relative prevalence of more than 5% at any passage are visualized. The plots are based on at least two independent cultures for each donor. Abbreviations: P, passage; CSA, cell subset panel A; CSB, cell subset panel B.

**Figure 5 cells-10-00218-f005:**
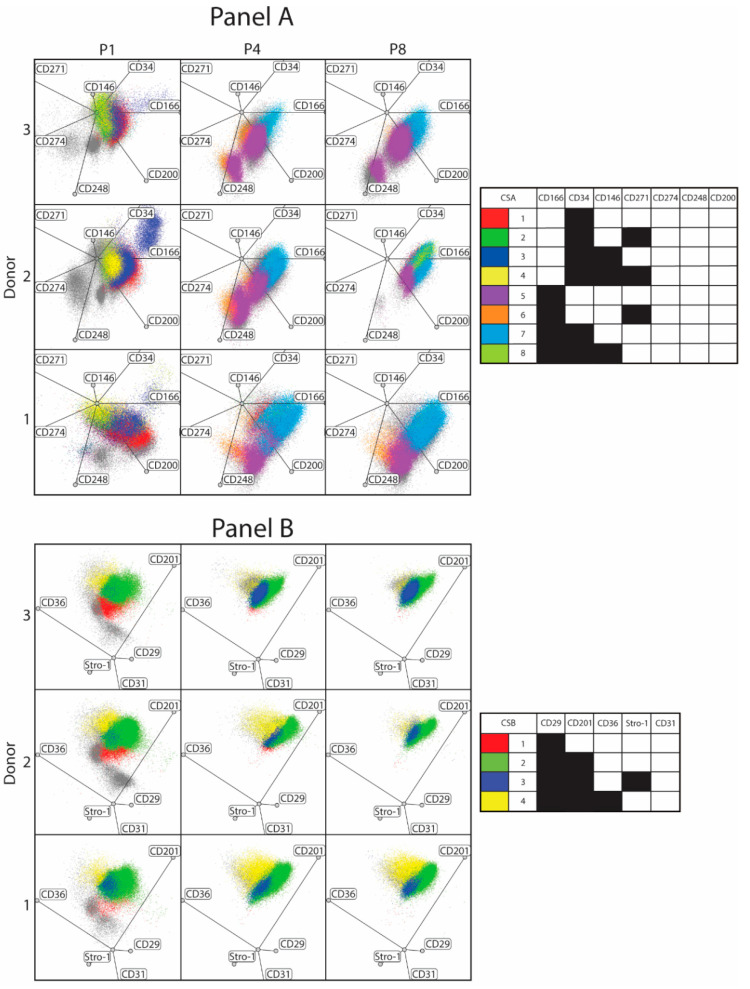
ASC subset clustering patterns rendered by multidimensional radar plots. The evolution of variant distribution is highly donor-specific, as is indicated by both the major and minor subpopulations. The plots are based on at least two independent cultures for each donor. Abbreviations: P, passage; CSA, cell subset panel A; CSB, cell subset panel B.

**Table 1 cells-10-00218-t001:** Antibody panel design for flow cytometry.

Laser	Channel	Dye	Panel A	Panel B
405 nm	450/45 BP	BV421	CD248	CD201
	525/40 BP	BV510	CD200	
	610/20 BP	BV605	CD166	CD36
	660/20 BP	BV650		CD29
561 nm	610/20 BP	PE-CF594	CD146	
	585/42 BP	FVS570	+	+
	780/60 BP	PE-Cy7	CD34	
638 nm	660/20 BP	Alexa Fluor 647		Stro-1
	712/25 BP	APC-R700	CD274	
	780/60 BP	APC-Cy7		CD31
488 nm	525/40 BP	BB515	CD271	

BP, bandpass; FVS570, fixable viability stain 570 was used in both panels.

**Table 2 cells-10-00218-t002:** The definition of the adipose-derived stem cell (ASC) subpopulation.

Panel	Subpopulation	Immunophenotype Profile
A	CSA1	CD166^–^**CD34^+^**CD146^–^CD271^–^CD274^–^CD248^–^CD200^–^
A	CSA2	CD166^–^**CD34^+^**CD146^–^**CD271^+^**CD274^–^CD248^–^CD200^–^
A	CSA3	CD166^–^**CD34^+^CD146^+^**CD271^–^CD274^–^CD248^–^CD200^–^
A	CSA4	CD166^–^**CD34^+^CD146^+^CD271^+^**CD274^–^CD248^–^CD200^–^
A	CSA5	**CD166^+^**CD34^–^CD146^–^CD271^–^CD274^–^CD248^–^CD200^–^
A	CSA6	**CD166^+^**CD34^–^CD146^–^**CD271^+^**CD274^–^CD248^–^CD200^–^
A	CSA7	**CD166^+^CD34^+^**CD146^–^CD271^–^CD274^–^CD248^–^CD200^–^
A	CSA8	**CD166^+^CD34^+^CD146^+^**CD271^–^CD274^–^CD248^–^CD200^–^
B	CSB1	**CD29^+^**CD201^–^CD36^–^ Stro-1^–^CD31^–^
B	CSB2	**CD29^+^CD201^+^**CD36^–^Stro-1^–^CD31^–^
B	CSB3	**CD29^+^CD201^+^**CD36^–^**Stro-1^+^**CD31^–^
B	CSB4	**CD29^+^CD201^+^CD36^+^**Stro-1^–^CD31^–^

CSA, cell subset panel A; CSB, cell subset panel B; boldface highlights positivity.

## Data Availability

Data are contained within the article.

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
