# Peer review of "Multiplex Analysis of Adipose-Derived Stem Cell (ASC) Immunophenotype Adaption to In Vitro Expansion"

_cells, 2021, doi:10.3390/cells10020218_

Round 1
Reviewer 1 Report
Review of “Multiplex analysis of adipose-derived stem cell (ASC) immunophenotype adaption to in vitro expansion” by Peng et al.
This paper describes some work that characterize changes in cell surface marker expression of primary adipose stem cells as they are cultured. This work is important and the studies reported here seem to be well done and timely. Most of my concerns have to do with how the results are interpreted.
Specific comments:
The last paragraph of the introduction appears to be written in a rather convoluted manner that needlessly obscures what was actually done. “…meaningfully represented clones” is not defined and can be interpreted multiple ways. What “higher level of complexity” is being talked about? “cell subset were resolved”, what does that mean in this context? A plainer and more direct description of the work accomplished would be helpful.
A significant part of the discussion talks about the link between expression of certain cell surface markers and function of these stem cells. While that work is clearly important, no functional assays were done in this work. It might also be appropriate to mention the limits of relying on cells surface marker expression alone in order to distinguish between cell types.
It seems to me that there are two possible alternative hypotheses as to why the expression changes with passage number, first that once cell type grows faster than another. The second possibility is that cells might change expression even without dividing. The authors should make some effort to explain why they see the changes that they do.
Finally, it is not new that cells grown in culture change. Much work has been done looking at alternate substrates or growing cells in 3D culture. If the goal of this work is to produce some sort of therapeutic device then it would seem those limitations of this work should at least be mentioned.
Reviewer 2 Report
The authors analyzed the evolution of defined ASC subsets from three donors in the course of eight tissue culture passages using multichromatic flow cytometry with two panels encompassing respectively seven and five cell surface markers. However, this reviewer is not convinced about the scientific interest of this work.
In particular:
- the study proposed is not original: other works already shown that ASC immunophenotype is subjected to variations during cell culture.
- it is not clear how the surface markers were chosen and distributed between the two panels
- the variability between different donors indicate that a wider number of subjects should be analyzed to strengthen the data
- the authors did not provide any explanation for the observed modulation of ASC phenotype during passages, and for the potential impact of this modulation on ASC therapeutic potential. This point should be better addressed.
Reviewer 3 Report
General Comments
The authors highlight the importance of evaluating the immunophenotype of the SVF and passaged ASC as a function of expansion. The authors provide novel observations regarding the co-expression of individual subsets of surface antigens in this context. The work is well performed albeit on a minimal sized study population of n = 3 donors which suggests that future investigation will be necessary on a larger cohort size. Several issues should be addressed:
- The authors have not exhaustively reviewed the relevant literature with respect to the surface antigen expression profile on ASC. For example, there have been earlier studies evaluating the presence of markers such as CD146 and CD166 (ALCAM) on ASC from over a decade ago. It would be appropriate to cite these studies.
- The authors study the donors at Passage 1, 4 and 8. In no case has the freshly isolated SVF cell population been examined as has been reported in prior studies. Why did the authors not include this element in the study? Understanding how the process of plastic adherence may influence surface immunophenotype would have been helpful.
- There is a body of literature describing the immunophenotype of bone marrow MSC that is often discussed in the context of ASC. The BMSC literature has historically not found evidence of CD34 expression and this aspect of the ASC surface immunophenotype has raised questions over the years. The authors should address this difference of the ASC immunophenotype relative to the BMSC in their Discussion or Introduction for sake of completeness and to highlight the relative novelty of their observations.
Specific Comments
Ln97. In addition to noting the population doublings, it would be informative to report the length of time for each passage period.
Discussion. The authors identify cluster of differentiation markers that are associated with temporal expression profiles defining the ASC populations. Since not all readers will be aware of the functionality of each CD protein, it would be helpful to identify the known function of the identified relevant CD markers such as CD29, CD34, CD166, CD201 in the context of the ASC and its potential utilization. Providing insights into how the markers may (or may not) provide understanding into cell function in the context of regenerative medicine and adipogenesis would be informative to readers who are not well versed in flow cytometry tools.
Discussion: The authors have limited their study to evaluation of 2D cultures only. It may be worth including a discussion of whether (or not) the expansion of the ASC in a 3D hydrogel scaffold would potentially influence the surface immunophenotype.
Incomplete references: 30, 32, 34, 36, 37.
Round 2
Reviewer 2 Report
The authors satisfactorily addressed most of the previous concerns.